



# Intercomparison of wind observations from ESA's satellite mission Aeolus, ERA5 reanalysis and radiosonde over China

Boming Liu[1], Jianping Guo[2*], Wei Gong[1*], Yong Zhang[3], Lijuan Shi[3], Yingying Ma[1], Jian Li[2], Xiaoran Guo[2], Ad Stoffelen[4], Gerrit de Leeuw[4], and Xiaofeng Xu[5]

[1] State Key Laboratory of Information Engineering in Surveying, Mapping and Remote Sensing (LIESMARS), Wuhan University, Wuhan, China
[2] The State Key Laboratory of Severe Weather, Chinese Academy of Meteorological Sciences, Beijing 100081, China
[3] Meteorological observation Centre, Chinese Meteorological Administration, Beijing 100081, China
[4] The Royal Netherlands Meteorological Institute (KNMI), 3730 AE De Bilt, The Netherlands
[5] Chinese Meteorological Administration, Beijing 100081, China

*Correspondence to*: Dr./Prof. Jianping Guo (Email: jpguocams@gmail.com) and Wei Gong (Email: weigong@whu.edu.cn)

**Abstract.** The European Space Agency (ESA) Earth Explorer Atmospheric Dynamics Mission Aeolus is the first satellite mission providing wind profile information on a global scale, and its wind products

have been released on 12 May 2020. In this study, we verify and intercompare the wind observations from ESA's satellite mission Aeolus and the European Centre for Medium-Range Weather Forecasts (ECMWF) fifth generation atmospheric reanalyses (ERA5) with radiosonde (RS) observations over China, to allow a fitting application of Aeolus winds. Aeolus provides wind observations in aerosol-free (referred to as Rayleigh-clear winds) and cloudy atmospheres (Mie-cloudy winds). In terms of

Aeolus and RS winds, the correlation coefficient (R) and mean difference of Rayleigh-clear (Mie-cloudy) vs RS winds are 0.90 (0.92) and 0.09±9.62 (−0.59±8.05) m/s, respectively. The vertical profiles of wind speed differences between Aeolus and RS winds are opposite to each other during ascending and descending orbits, indicating that the performance of Aeolus wind product is affected by the orbit phase. The comparison of ECMWF winds relative to Aeolus winds provides the R and

mean difference of Rayleigh-clear (Mie-cloudy) winds, which are 0.95 (0.97) and −0.16±6.78 (−0.21±3.91) m/s, respectively. The Rayleigh-clear and Mie-cloudy winds are almost consistent with the ECMWF winds, likely due to the assimilation of Aeolus wind observations into the ECMWF winds.

Moreover, we find that among the results of comparing Aeolus with RS and ECMWF winds, the wind speed difference of Rayleigh-clear winds is large in the height range of 0−1 km, especially during descending orbits. This indicates that the performance of low-altitude Rayleigh-clear wind products could be affected by the near-surface aerosols. In addition, the R and mean difference between ERA5

and RS zonal wind components are 0.89 and −1.46±6.33 m/s, respectively. The RS zonal winds tend to be larger than those from ERA5. The wind speed difference between RS and ERA5 zonal winds in low-lying area is low and insignificant, while it is relatively high and significant over the Qinghai-Tibet Plateau areas. Overall, the Aeolus winds over China are similar to the RS and ECMWF winds. The RS and ERA5 zonal winds are somewhat different over high altitude area, but these differences

are acceptable for application of wind products. The findings give us sufficient confidence and information to apply Aeolus wind products in numerical weather prediction in China and in climate change research.

# 1 Introduction

Atmospheric three-dimensional wind fields play a key role for the prediction of extreme events (Pu et

al., 2010; Guo et al., 2018; Stettner et al., 2019), a better understanding of air pollution dispersion (Liu et al., 2018; Yang et al., 2019; Shi et al., 2020; Su et al., 2020; Zhang et al., 2020) and complex aerosol-cloud-precipitation interactions (Koren et al., 2005; Fan et al., 2009; Li et al., 2011; Guo et al., 2017; Liu et al., 2020a). Moreover, assimilating atmospheric wind observations into numerical weather prediction models is of great significance for better predicting surface wind speed and assessing the

trend of wind energy changes (Benjamin et al., 2004; Constantinescu et al., 2009; Simonin et al., 2014). Therefore, continuous global wind profile observations are desperately needed for advancing our knowledge of atmospheric dynamics as well as for improving the accuracy of numerical weather prediction (Houchi et al., 2010; Albertema et al., 2019; Stoffelen et al., 2005; 2020).



The European Space Agency (ESA) Earth Explorer Atmospheric Dynamics Mission Aeolus, launched on 22 August 2018, is the first satellite mission designed to acquire atmospheric wind profiles on a global scale by employing a Doppler Wind Lidar payload, called Atmospheric LAser Doppler INstrument (ALADIN, Reitebuch, 2012; ESA, 2008; Stoffelen et al., 2005). The flight altitude and

repeat cycle of the Aeolus satellite are about 320 km and 7days, respectively (Witschas et al., 2020). It provides measurements of the wind vector component along the instrument's line-of-sight from the ground up to 30 km altitude (Rennie et al., 2020). The Aeolus wind products have been released on 12 May 2020 (https://earth.esa.int/eogateway/news/aeolus-data-now-publicly-available, last access: 13-01-2021), the vertical resolution is 0.25 to 2 km, while the wind accuracy is approximately 2 to 4 m/s,

depending on the altitude (Tan et al., 2017; Guo et al., 2020). The most unique products are Aeolus L2B wind products, which include the geo-located and consolidated horizontal line-of-sight (HLOS) wind observations with actual atmospheric correction and bias corrections applied. It provides scientific wind products for public users, which contain the Rayleigh-clear and Mie-cloudy winds (Tan et al., 2008). The Rayleigh-clear and Mie-cloudy winds refer to wind observations in an aerosol-free

or a cloudy atmosphere, respectively (Witschas et al., 2020). The European Centre for Medium-Range Weather Forecasts (ECMWF) started to assimilate Aeolus data in their operational forecasts as of January 2020, using an internal bias correction based on their model winds (https://earth.esa.int/eogateway/news/aeolus-winds-now-in-daily-weather-forecasts, last access: 15-01-2021).

To date, the Aeolus team and related researchers across the world have carried out a series of verification and comparison work on the Aeolus wind products (Marksteiner et al., 2018; Rennie et al., 2018; Kanitz et al., 2020). Rennie and Isaksen (2020) conducted the first validation of the Aeolus Level 2B product against the ECMWF numerical weather prediction model, which played a crucial role in the Aeolus characterization. At the same time, Witschas et al. (2020) carried out comprehensive

validation of Aeolus measurements against airborne Doppler wind lidar observations. The mean biases





of line-of-sight wind speeds between the airborne Doppler wind lidar and the Aeolus were −0.9 m/s and +1.6 m/s, respectively (Lux et al., 2020). Moreover, the Aeolus wind products have been compared with RS data above the Atlantic Ocean, from which the systematic and statistical errors of the Rayleigh (Mie) winds were found less than 1.5 (1.0) and 3.3 (1.0) m/s, respectively (Baars et al., 2020). Martin

et al. (2020) compared the Aeolus winds with radiosonde (RS) observations and two numerical weather prediction model equivalents, confirming that the performance of Aeolus wind products can be easily affected by satellite flight direction, seasonal differences and geographic changes. In addition, Guo et al. (2020) evaluated the performance of Aeolus wind products over China by comparing with data from the ground-based radar wind profiler network of China. The above-mentioned verification

exercises have deepened our understanding of the global Aeolus wind products and led to improvements in Aeolus products. However, since the radar wind profiler network of China is mainly distributed in eastern China, the performance of Aeolus wind products over central and western parts of China has not been specifically investigated. To fill this gap, it is worthwhile to further verify the performance of Aeolus wind products in China using the China's RS observation network. This

network comprises 120 RS sites homogeneously distributed across mainland China and provides atmospheric soundings of winds (Guo et al., 2016; 2019). The use of these data, together with ERA-5 reanalysis data, provides a unique opportunity to deepen our understanding of the performance of Aelous wind products over all mainland China.

To this end, a comparison is made of Aeolus, ERA5 and RS wind data over China. First, we analysed

the correlations and differences between each of the three data sets, and then spatially investigated their wind component differences vertically and horizontally. The paper is organized as follows. First, the three types of wind data such as Aeolus, ERA5 and RS are briefly described (Section 2.1-2.3); the data matching principles are represented in detail in Section 2.4; and the statistical approach is addressed in Section 2.5. Section 3 presents a comprehensive comparison of Aeolus, ERA5 and RS

wind observations. Finally, the main findings are summarized in Section 4.





## 2 Data and methods

### 2.1 Aeolus wind data

Aeolus is the first ever satellite designed to directly observe atmospheric wind profiles on a global scale (ESA, 2008; 2016). The ground tracks of Aeolus over China are shown in Figure 1. The red and

blue lines represent the ascending and descending ground tracks, respectively. In this study, the Aeolus Level 2B (L2B) wind products from 20 April 2020 to 30 September 2020 are collected for comparison. The L2B products contain the HLOS winds for the Rayleigh-clear winds and Mie-cloudy winds. The quality of the Aeolus wind data is indicated by validity flags (0=invalid and 1=valid) and estimated errors. The estimated error is the theoretical value calculated based on the signal intensity as well as

the temperature and pressure sensitivity of the Rayleigh channel response (Dabas et al., 2008), which can be used as data filtering threshold (discussed in section 2.4). In addition, the orbit of Aeolus includes the ascending (HLOS points east) at 00:00 UTC and descending (HLOS points west) orbits at 12:00 UTC. More detailed descriptions of Aeolus wind products are provided in previous studies (Reitebuch et al., 2012; ESA, 2016; Rennie et al., 2018).

### 2.2 Radiosonde wind data

The L-band RS is a new-generation sounding system that can provide fine-resolution profiles of atmospheric parameters (Guo et al., 2016). Its principle is to carry the radiosonde to high altitude through balloons to measure meteorological parameters such as temperature, pressure, humidity, wind direction and wind speed (Liu et al., 2019). Moreover, it also records the sampling time, altitude and

geographic coordinates of each sampling point. The RS in China are launched twice per day at around 0000 and 1200 UTC (Zhang et al., 2020). The spatial distribution of the RS stations used in this study is shown in Figure. 1, indicated by the yellow dots. The green line approximates 90E, which is the approximate position of Aeolus at 12:00 and 0:00 UTC. Note that at the launch times, RS in eastern



China are taken about 2.5 hours after the Aeolus overpass time. This is, if at all, eastern RS will mostly match with later Aeolus tracks slightly to the west of the RS station. In this study, the RS wind data over China from 20 April 2020 to 30 September 2020 were collected for comparison. It is noted that, because the access permission of RS data was updated in June 2020, the geographic coordinates of

each individual sampling point in the RS profile were only available from 20 April 2020 to 31 May 2020.

*2.3 ERA5 wind data*

The ECMWF produces global numerical weather predictions for Member and Co-operating States and reanalysis data for the broader community (Hoffmann et al., 2019; Belmonte et al., 2019). The fifth

generation ECMWF atmospheric reanalysis system (ERA-5) provides hourly wind fields on a 0.25 x 0.25-degree latitude/longitude grid (Hersbach et al., 2020). The zonal, u, and meridional, v, component of wind can be download from https://cds.climate.copernicus.eu (last access: 13-01-2021), and the speed and direction of the horizontal wind can be calculated from the u and v component of wind (Liu et al., 2020b). The ERA5 data over China from 20 April 2020 to 30 September 2020 were download

for the comparison with RS data.  Note that the ERA5 data has a total of 28 layers in the height range of 0–20 km. It is not appropriate to use the value of a certain layer to match the corresponding Aeolus bin value. Therefore, following the study of Lux et al. (2020), the operational ECMWF wind data from the Aeolus Level 2C (L2C) wind product are obtained to compared with Aeolus winds. The Aeolus L2C wind product contains the wind field results from the ECMWF operational prediction model

assisted by Aeolus. It provides the corresponding HLOS wind, u and v components of the ECMWF wind vector for each Aeolus bin. Due to the Aeolus L2C wind product is first published on 9 July 2020. Therefore, the Aeolus L2C products from 9 July 2020 to 30 September 2020 are also collected to provide the ECMWF HLOS winds. In addition, the difference between the ERA5 and ECMWF wind data is likely caused by the fact that the EAR5 dataset is entirely independent of the Aeolus winds

(https://confluence.ecmwf.int/display/CKB/ERA5%3A+data+documentation#ERA5:datadocumentat



ion-Observations: last access: 15-01-2021), but the ECMWF winds are a compromise between assimilated Aeolus winds and the ECMWF short-range forecast (background) (https://earth.esa.int/eogateway/news/aeolus-winds-now-in-daily-weather-forecasts: last access: 15-01-2021). Although the ERA5 model is not identical to the ECMWF model used in operation, its upper
5   air characteristics are very similar, since ECMWF used a recent model cycle for ERA5, but runs it at a slightly lower resolution than that in operation.

*2.4 Data matching procedures*

Due to the quite different spatial-temporal resolutions and sampling of Aeolus, RS and ERA5 data, data matching procedures have to be defined before performing comparison analysis. The
10  corresponding flowchart is presented in Figure 2, and its detailed procedures are presented below.

*2.4.1 Aeolus and RS data*

For RS and Aeolus data, the RS winds firstly should be converted to Aeolus vertical bin. The RS wind vector in each bin is projected onto the Aeolus HLOS wind using the following equation (Witschas et al., 2020):

$$v_{RS_{HLOS}} = \cos(\psi_{Aeolus} - WD_{RS}) \cdot WS_{RS} \tag{1}$$

where $\psi_{Aeolus}$ represents the Aeolus azimuth angle; $WS_{RS}$ and $WD_{RS}$ are the RS wind speed and direction, respectively. Moreover, for the Aeolus bins, the height interval of each bin should be covered by RS measurements, and the average of $v_{RS_{HLOS}}$ in each Aeolus vertical bin are taken in attempt to compare with the corresponding Aeolus HLOS winds.

20  Next, for Aeolus winds, the Aeolus official documentation pointed out that the validity flag and estimated errors should be used to conduct quality control (Dabas et al., 2008; Tan et al., 2017). A sensitivity analysis on the estimated errors can be seen in Figure S1. The difference between Mie-



cloudy (Rayleigh-clear) wind and RS wind is large when the estimated error is larger than 6 (10) m/s. Therefore, following previous studies (Guo et al., 2020; Witschas et al., 2020), the Aeolus wind data are accepted only when the validity flag equals 1 and the estimated errors for wind are, respectively, less than 7 and 5 m/s for Rayleigh and Mie channels.

Finally, we pose constraints on the time and geographical matches before comparison. Specifically, the time difference between the RS and Aeolus wind profiles should be minimum. Due to the fact that there are only two RS wind profiles (i.e., at 00:00 and 12:00 UTC) every day for a given site, setting the time difference threshold too small will result in a small number of matching stations; see Figure 1. Therefore, the time difference between the RS and Aeolus wind profiles is set within one hour. Note

that 1 hour corresponds to an earth rotation of 15 degrees, i.e., a maximum distance to the Aeolus track of about 1500 km. Meanwhile, following the well-established geographical matching principle (Guo et al., 2020), the distance between an Aeolus wind profile and an RS site should be less than 75 km. After temporal and spatial collocation, the closest Aeolus wind profile to each RS profile is adopted for comparison.

In addition, it should be noted that RS has the problem of time and space drift in measuring the vertical wind profile (Baars et al., 2020; Martin et al., 2020). As shown in Fig. 3, there are some time and space differences between each pair of samples in collocating the profiles, due to the extended measurement cycle of a RS profile. Figs. 3a and 3b show, as an example, the distance and time difference between Aeolus and a RS wind profile, respectively. Although we use the start coordinates and start time of the

RS to conduct the time and space matching with Aeolus data, the drift of a RS may break these matching rules, especially at high altitude (above 15 km). It is not certain whether the drift of the RS will have a significant impact on the comparison results. Therefore, we analyse the impact of distance and time mismatches on the differences between Aeolus and RS wind speed, as shown in Figure 4. Note that the data used here are collected from 20 April 2020 to 31 May 2020, where the geographic

coordinates of each sampling point in the RS profile are present (Section 2.2). Moreover, these data

have also been screened for validity and estimated error thresholds. The blue and red boxes represent the Mie-cloudy and Rayleigh-clear winds, respectively. The text labels represent the mean of the differences between the Aeolus and RS HLOS wind speeds in the corresponding box. The box size shows the upper margin, upper quartile, median, lower quartile, lower margin of the difference in each

bin. For Mie-cloudy winds (Figs. 4a and 4b), the difference between Mie-cloudy wind and RS winds is relatively constant with the distance and time differences increased. Similar to Mie-cloudy wind, the difference between Rayleigh-clear wind and RS wind is constant with the distance and time differences increased (Fig. 4c and 4d). These results indicate that the drift of the RS has rather small effect on the data that we use for comparison, the impact of RS drift can be excluded on subsequent comparisons.

Note that Mie-cloudy has a tendency of larger dispersion of differences for close matches in time and space. Close matches in time and space are mainly over western China, close to the yellow line in Figure 1. The difference may be due to geographical area, rather than due to time and space separation.

*2.4.2 Aeolus and ECMWF data*

For the comparison of Aeolus and ECMWF data, the Aeolus L2C wind product contains the HLOS

wind of ECMWF wind vector at each Aeolus bin. It means that we can obtain these two sets of data directly based on the Aeolus bin. The data for comparison is then filtered based on validity flag and the estimated errors. Note that the Aeolus L2C wind product is first published on 9 July 2020. Therefore, the matching period of Aeolus and ECMWF data is from 9 July 2020 to 30 September 2020.

*2.4.3 RS and ERA5 data*

In order to get more samples, the RS profile is directly matched with the gridded ERA5 data. The matching principle is to ensure that the time and space differences are minimized. Because of the daily launch time of RS twice per day at around 00:00 and 12:00 UTC, the gridded (0.25 x 0.25 degrees) ERA5 data at 00:00 and 12:00 UTC were downloaded. Meanwhile, according to the RS site coordinates, the ERA5 wind profile on the nearest grid position is obtained to match the RS wind

profile. In the vertical direction, the ERA5 data has a total of 28 layers in the range of 0-20 km. For each layer, the data were matched with the closest sample point on the RS profile according to the pressure coordinates. Note that to be more compatible and comparable with the HLOS statistics, the RS winds are converted to zonal components to compare with ERA5 zonal components. In addition,

the impact of RS drift on the difference between RS and ERA5 zonal wind components was also analysed as shown in Figure S2. Similarly, the data in Figure S2 show that the effect of RS drift can be ignored after the matching process.

*2.5 Statistical method*

The wind component difference between wind component sample A and wind component sample B is
given by:

$$v_{diff} = v_A - v_B \tag{2}$$

Moreover, the mean difference (MD) and standard deviation (SD) of $v_{diff}$ for n samples are estimated according to:

$$\text{MD} = \frac{1}{n} \sum_{i=1}^{n} v_{diff} \tag{3}$$

and

$$\text{SD} = \sqrt{\frac{1}{n-1} \sum_{i=1}^{n} (v_{diff} - MD)^2} \tag{4}$$

The correlation coefficient (R) between wind component sample A and wind component sample B is calculated using:

$$\text{R} = \frac{\sum_{i=1}^{n}(x_i - \bar{x})(y_i - \bar{y})}{\sqrt{\sum_{i=1}^{n}(x_i - \bar{x})^2}\sqrt{\sum_{i=1}^{n}(y_i - \bar{y})^2}} \tag{5}$$

where $x_i$ and $y_i$ represent the i-th sample point of A and B wind component dataset, respectively. $\bar{x}$ and $\bar{y}$ represent the mean wind component of the A and B wind component datasets, respectively.



# 3 Results and discussion

## 3.1 Overall intercomparison

The results of the comparison between Aeolus, RS and ERA5 (ECMWF) are shown in Figure 5. The red and blue dots represent the Rayleigh-clear and Mie-cloudy winds, respectively. Figure 5a shows the scatter plots of Aeolus against RS wind HLOS. The numbers of data pairs for the Rayleigh-clear and Mie-cloudy winds against RS winds are 12,205 and 1,328, respectively. The slopes of the linear fits and the values of R for the Rayleigh-clear (Mie-cloudy) vs RS winds are 0.92 (0.78) and 0.90 (0.92), respectively. The mean difference of RS HLOS relative to Rayleigh-clear and Mie-cloudy HLOS are $0.09\pm9.62$ and $-0.59\pm8.05$ m/s, respectively (Fig. 5d). These results indicate that the Aeolus Rayleigh-clear and Mie-cloudy wind products over China are on average consistent with RS wind observations, but with large dispersion. Figure panels 5b and 5e illustrate the comparison between the Aeolus and ECMWF HLOS. The numbers of data pairs are 135,393 and 77,694 for Rayleigh-clear and Mie-cloudy winds, respectively. The slopes of the linear fits (R) are 0.90 (0.95) and 0.96 (0.97) for Rayleigh-clear and Mie-cloudy wind speed against ECMWF wind speed, respectively. The mean difference is $-0.16\pm6.78$ and $-0.21\pm3.91$ m/s for ECMWF winds relative to Rayleigh-clear and Mie-cloudy winds, respectively. As expected, since the HLOS observation data of Aeolus were assimilated in ECMWF operations, the wind values of Aeolus and ECMWF agree well. As a matter of fact, the comparison results between Aeolus and ECMWF winds depend (only) on the data assimilation settings. Figure panels 5c and 5f show the comparison results between RS and ERA5 zonal wind components. The sample numbers, slopes of linear fits, values of R and mean difference of RS vs ERA5 zonal winds are 127,927, 0.92, 0.89 and $-1.46\pm6.33$ m/s, respectively..

The spatial distributions of the correlation coefficients between the Aeolus, RS and ERA5 (ECMWF) wind speed data set are shown in Figure 6. In addition, the spatial distributions of the number of paired data samples is shown in Figure S3. Figure panels 6a and 6b show the comparison results between



Aeolus and RS HLOS winds. For Mie-cloudy winds, the correlation coefficients of most sites (41 out of 54, i.e., 76 %) are exceeding 0.8, lower values are observed at some sites on Qinghai-Tibet Plateau. Similar to Mie-cloudy winds, also for the Rayleigh-clear winds, the R at most of the sites (36 out of 54, i.e., 67 %) is high and these sites are located in the low-lying area, while the R value of the sites located in the Qinghai-Tibet Plateau area are lower than 0.8. This phenomenon also occurs in the comparison results between RS and ERA5 zonal wind components (Fig. 6e). The sites with high R are located in the low-lying area, while the sites with low R are mainly located in the Qinghai-Tibet Plateau area. As for the comparison result of Aeolus and ECMWF winds, we can see that the correlation between Rayleigh-clear (Mie-cloudy) and ECMWF winds along the ground track is very large (Fig. 6c and 6d). This result again indicates the consistency of Aeolus winds and ECMWF wind speeds, mainly due to the assimilation of the Aeolus observation data. Overall, the performance of Aeolus, RS and ERA5 wind products over China is similar to each other.

*3.2 Vertical distribution of wind differences*

To investigate the wind differences in the vertical, we analysed the variations of Aeolus HLOS winds and RS zonal winds at three different heights (850, 500 and 100 hPa), using the ERA5 zonal wind data as the background values. The spatial distributions of averaged Aeolus HLOS winds, RS zonal winds and ERA5 zonal winds at the different height are presented in Fig. 7. Note that for Aeolus Rayleigh-clear and Mie-cloudy winds, the averaged value result is only provided when the number of valid samples in a bin is larger than 10. The average HLOS is calculated from all valid HLOS at the corresponding height and during the observation period. At 850 and 500 hPa (approximately 1.3 and 5.5 km above ground level), the average HLOS of Rayleigh-clear, Mie-cloudy and RS are similar to that of ERA5, except in the Qinghai-Tibet Plateau and Yunnan areas where the results of RS zonal winds are relatively high (Figs. 7g and 7h). Compared with the average HLOS at 850 and 500 hPa, it is found that the average HLOS at 100 hPa (approximately 15 km above ground level) are generally larger. Because of the stratosphere is mainly atmospheric molecules (clouds occasionally appear), the





number of valid samples of Rayleigh-clear winds has increased significantly. At this level, most of average HLOS results of Mie-cloudy winds are consistent with that of ERA5 (Fig. 7c). Moreover, for Rayleigh-clear winds (Fig. 7f), most of the average HLOS results are similar to that of ERA5. By contrast, as for RS zonal winds (Fig. 7i), the averaged zonal winds are similar to that of ERA5 over

China except in the Qinghai-Tibet Plateau area, though this is not directly comparable to HLOS wind component biases.

The vertical distributions of the wind speed differences between Aeolus, RS and ERA5 (ECMWF) for ascending and descending obits are shown in Figure 8. For the comparison of Aeolus and RS HLOS data, the mean wind difference between the Mie-cloudy (Rayleigh-clear) and the RS winds is less than

2 m/s in the height range of 1−15 km during all observation times. During ascending and descending orbits, the vertical distributions of the HLOS differences are opposite to each other, as one may expect from the opposite sense of the HLOS. The maximum mean HLOS difference of the Mie-cloudy and Rayleigh-clear relative to RS winds during ascending (descending) orbits is −5.12±5.49 (7.06±9.26) and 5.44±9.12 (−5.76±10.26) m/s, respectively. In principle, this could be due both to ECMWF model

zonal bias or Aeolus orbit-phase dependent bias. This may be further investigated using the RS winds. For Aeolus and ECMWF data (Figs. 8d, 8e and 8f), due to the Aeolus assimilation at ECMWF, the HLOS difference between Aeolus and ECMWF along the vertical profile is small. At the same time, it is noted that two similar phenomena can be found in the comparison results of Aeolus with RS and ECMWF. The first observation is that the wind speed difference of Rayleigh-clear winds is large in

the height range of 0−1 km, which may be due to limited performance of Rayleigh-clear products at low received power. On the other hand, at the height range of 0−1 km, the difference of Rayleigh-clear winds during descending orbits is larger than that during ascending orbits. This may be caused by the diurnal variation of aerosols in the atmospheric boundary layer. At descending time (approximately 1800 LT=UTC+6), the aerosol layer can be elevated to approximately 1–2 km, in

which the strong aerosol scattering in the boundary layer would inevitably contaminate the molecular



scattering signal, thereby reducing the inversion accuracy of Rayleigh wind from Aeolus (Tan et al., 2017). This result is similar to that in the previous validation work (Guo et al., 2020). For the comparison of RS and ERA5 zonal wind components (Figs. 8g, 8h and 8i), the vertical distributions of the wind differences are similar during all time, ascending and descending orbits. The RS zonal

winds tend to be larger than ERA5 zonal winds in the height range of 1-11 km, while less than that in the height range of 11-16 km. If winds over China were generally zonal, then this speed bias would bias ERA5 HLOS low with respect to Aeolus in the height range of 1-11 km for ascending orbits (HLOS pointing east) and high for descending orbits (HLOS pointing west). It is unclear why the Aeolus bias follows this bias pattern against the radiosondes. Overall, the three types of wind data have

certain differences along the vertical profile, which need further investigation.

*3.3 Horizontal distribution of wind differences*

To investigate the spatial distribution of wind difference, the variations of Aeolus HLOS winds, RS zonal winds and ERA5 zonal winds during ascending and descending orbits are investigated. The spatial distributions of the averaged value of Aeolus HLOS winds and RS zonal winds during

ascending and descending orbits are presented in Figure 9, where the ERA5 zonal wind data is used as the background values. The average HLOS result is only provided when the number of valid samples for Aeolus Rayleigh-clear and Mie-cloudy winds at each bin is larger than 10. The average wind is calculated from all valid winds on the Aeolus or RS profile during the corresponding observation period. During ascending orbits, the average winds of Rayleigh-clear and Mie-cloudy are similar to

those of ERA5, with the average HLOS wind is higher in northern China and lower in the south China. The spatial distribution of average zonal winds of RS follows a similar pattern as that of ERA5, but the average zonal winds of RS is slightly larger (Fig. 9e). Similarly, during descending orbits, the spatial distributions of the average winds of Rayleigh-clear, Mie-cloudy and RS are similar to that of ERA5, but there is a difference in the magnitude of the average winds. To quantify these wind

differences, their spatial distributions of these wind differences are investigated.





The spatial distribution of wind difference between Aeolus, RS and ERA5 (ECMWF) during the ascending orbits are shown in Figure 10. For the comparison of Mie-cloudy and RS winds, the average HLOS difference of most the sites (25 out of 32, i.e., 78 %) is negative, and the mean difference for all sites is −2.18±4.17 m/s, indicating the Mie-cloudy HLOS winds are smaller than RS HLOS winds during ascending orbits. In contrast, for the Rayleigh-clear winds, the averaged wind HLOS difference is positive at half of the sites (17 out of 33, i.e., 52 %), and the mean difference for all sites is 0.77±2.29 m/s. The wind HLOS differences of Mie-cloudy and Rayleigh-clear relative to ECMWF are presented in Figure 10c and 10d. We note that the average wind HLOS difference of almost all sites is less than 2 m/s, and the mean difference for Mie-cloudy and Rayleigh-clear HLOS winds is −0.17±0.99 and 0.05±1.11 m/s, respectively. For the comparison of RS and ERA5 zonal wind components, the averaged wind speed difference is positive at all sites, and the mean difference for all sites is 2.30±1.56 m/s, indicating that the RS zonal winds are larger than ERA5 zonal winds during ascending orbits.

For the descending orbits, the wind speed differences between Aeolus, RS and ERA5 (ECMWF) are presented in Figure 11. Opposite to the ascending orbits, the average wind HLOS difference between Mie-cloudy and RS winds is positive at most sites (30 out of 38, i.e., 79 %) during descending orbits, and the mean difference for all sites is 2.75±5.13 m/s. For the comparison of Rayleigh-clear and RS winds, the averaged wind HLOS difference of half sites (22 out of 40, i.e., 55 %) is negative, and the mean difference for all sites is  −0.65±2.39 m/s. Similar to the ascending orbits, the average wind HLOS difference between Aeolus and ECMWF winds is less than 2 m/s at almost all sites, and the mean difference for Mie-cloudy and Rayleigh-clear winds is 0.45±0.89 and 0.17±1.03 m/s, respectively. Moreover, for the comparison of RS and ERA5 zonal wind components during descending orbits, the averaged wind speed difference of all sites is also positive, and the mean difference for all sites is 2.26±1.60 m/s, which means that the RS zonal winds are also larger than ERA5 zonal winds during descending orbits.



The above results indicate that the change of observation time has an impact on the spatial distribution of wind speed difference between the Aeolus and RS HLOS winds. One reason is that the Aeolus field of view direction points to the right of the spacecraft into the dark side of the earth, implying that the viewing direction during the ascending (HLOS points east) and descending (HLOS points west) orbits are different. Another reason may be that the meteorological conditions such as wind speeds, BLH, air temperature and aerosol distributions during the ascending and descending orbits are different (Sun et al., 2014), though the ERA5 maps are very similar for ascending (00:00 UTC) and descending (12:00 UTC). In addition, from the comparison results of RS and ERA5 zonal winds, we note that the average wind difference between RS and ERA5 zonal winds in low-lying areas is low and insignificant, while it is relatively high and significant in high-altitude areas. Overall, the Aeolus HLOS winds are similar to the RS (ECMWF) winds during all observation times, and the mean differences for Mie-cloudy and Rayleigh-clear winds are $0.91 \pm 4.96$ ($0.19 \pm 0.89$) and $0.03 \pm 2.27$ ($0.13 \pm 1.01$) m/s, respectively (Fig. S4). The RS and ERA5 zonal winds are somewhat different ($2.28 \pm 1.56$ m/s), but these differences are acceptable for application of wind products.

## 4 Summary and conclusions

This study presents a comprehensive intercomparison analysis between the wind observations from Aeolus, ERA5 model winds and radiosondes (RS) over China during the period 20 April 2020 to 30 September 2020. The correlation and difference between each set of data have been analysed to evaluate the Aeolus wind product characteristics. Furthermore, the spatial vertical and horizontal distributions of wind differences between each set of data have been investigated.

First, for the Aeolus and RS HLOS winds, the R and mean difference between the Rayleigh-clear (Mie-cloudy) and RS winds are 0.90 (0.92) and $0.09 \pm 9.62$ ($-0.59 \pm 8.05$) m/s, respectively. The vertical profiles of the HLOS differences are opposite to each other during ascending and descending orbits.





Moreover, for the Mie-cloudy winds, the average HLOS differences are negative during ascending orbits (−2.18±4.17 m/s), whereas they are positive during descending orbits (2.75±5.13 m/s). These results indicate that the performance of Aeolus wind products may be affected by the orbit phase or HLOS wind conditions. For the Aeolus and ECMWF winds, as expected, due to assimilation of the

Aeolus wind products in the ECMWF model, the wind results of Aeolus and ECMWF show good consistency spatially. The R and mean difference of Rayleigh-clear and Mie-cloudy HLOS winds relative to ECMWF winds are 0.95 (0.97) and −0.16±6.78 (−0.21±3.91) m/s, respectively. The correlation and difference between Aeolus and ECMWF winds mainly depend on the data assimilation settings. In addition, the wind HLOS difference of Rayleigh-clear winds relative to RS or ECMWF

winds are large in the height range of 0−1 km, especially during descending orbits. This result indicates the possible influence of surface aerosol contamination on the performance of Rayleigh-clear wind products. For the RS and ERA5 zonal winds, the mean difference and R of ERA5 and RS zonal winds are −1.46±6.33 m/s and 0.89, respectively. Along the vertical, the wind speed differences are positive in the height range of 0-11 km and negative in the height range of 11-16 km. Spatially the

wind difference of RS versus ERA5 zonal winds in low-lying areas is low and insignificant, while is relatively high and significant in high-altitude areas. These results indicate that the RS zonal winds tend to be higher than those of ERA5.

Our work comprehensively compares the wind products form Aeolus, RS and ERA5 (ECMWF) over large regional spatial scales in mainland China. From the perspective of correlation and magnitude of

difference between samples, we conclude that the winds form Aeolus, RS and ERA5 (ECMWF) are similar to each other. Some profile biases warrant further investigation however. The information collected is helpful to improve our confidence of applying Aeolus wind products in numerical weather prediction over China and in research on climate change.

### Data availability

The L-band radiosonde data used in this paper can be provided for non-commercial research purposes upon request (Dr. Jianping Guo: jpguocams@gmail.com). The Aeolus dataset can be downloaded from https://aeolus-ds.eo.esa.int/oads/access/collection (last accessed 24 October 2020). Instructions for use

and data access methods can be found on the official website. The ERA5 wind data can be download from https://cds.climate.copernicus.eu/cdsapp#!/dataset/reanalysis-era5-pressure-levels?tab=form.

### Author contributions

The study was completed with close cooperation between all authors. J. Guo and B. Liu conceived of

the idea for assessing the Aeolus wind products using radiosonde measurements in China; J. Guo and B. Liu conducted the data analyses and co-wrote the manuscript; Y. Ma, W. Gong, A. Stoffelen, G. Leeuw, and X. Xu discussed the experimental results, and all coauthors helped reviewing the manuscript.

### Competing interests.

The authors declare that they have no conflict of interest.

### Special issue statement.

This article is part of the special issue "Aeolus data and their application". It is not associated with a conference.

### Acknowledgements.

We are very grateful to the China Meteorological Administration for the operation and maintenance of the radiosonde observational network, to ESA for the Aeolus data and to ECMWF for ERA5. This work was financially supported by the National Natural Science Foundation of China under grants 42001291, 91637211 and 41771399. The study contributes to the ESA/MOST cooperation project DRAGON5, Topic 3 Atmosphere, sub-topic 3.2 Air-Quality and the ESA Aeolus DISC project.



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





**Figures:**

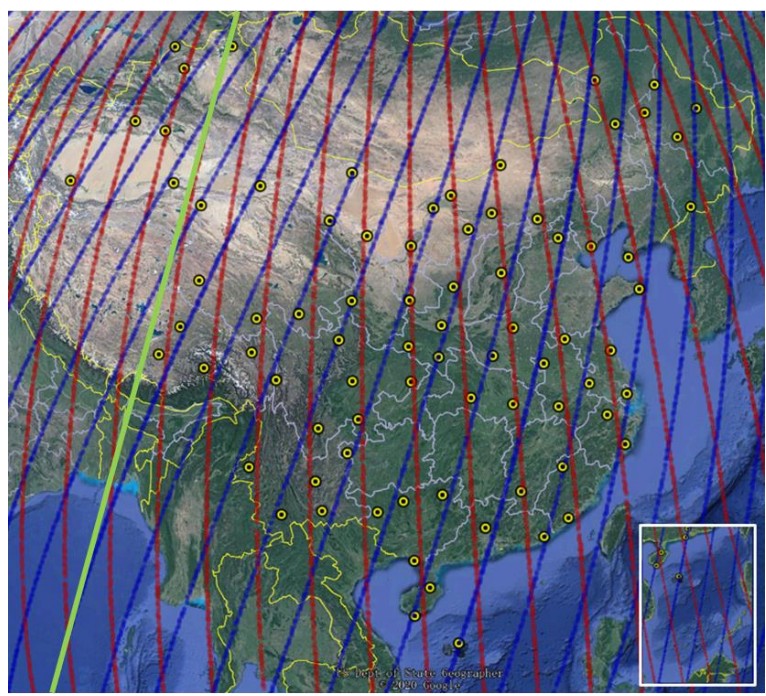

**Figure 1.** Geographic distribution of RS sites and Aeolus ground tracks superimposed on the GoogleEarth map of China (© Google Maps). Red and blue lines represent the Aeolus ground tracks for ascending and descending orbits, respectively. The yellow dots denote the RS sites. The green line approximates 90E, which is the approximate position of Aeolus at 12:00 and 0:00 UTC, the launch

10    times of the radiosondes. Eastern RS are taken about 2.5 hours after Aeolus overpass time.





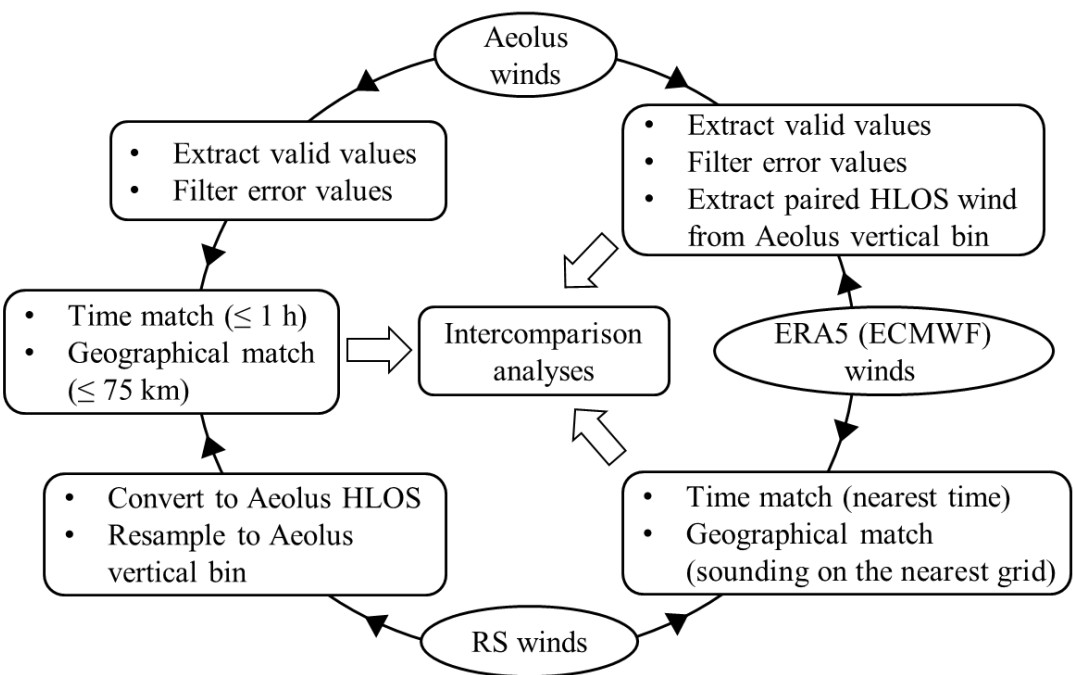

**Figure 2.** Flowchart showing the main processing procedures used to compare the wind observations from Aeolus, RS and ERA5 (ECMWF).





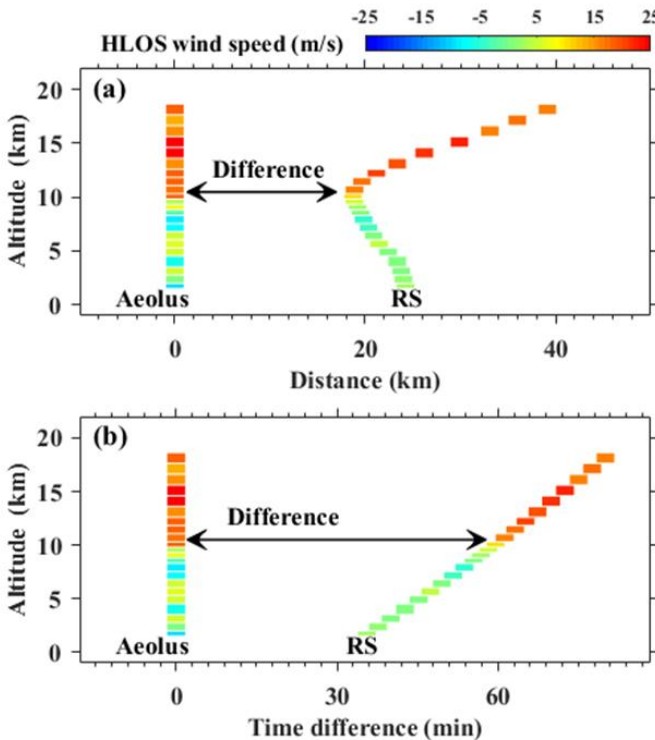

**Figure 3.** The examples on 24 Apr 2020 illustrating how the Aeolus HLOS profile is matched with RS HLOS profile in terms of (a) distance and (b) time difference between paired samples, which is based on the collocated time and space of Aeolus sampling grid as described in the text.



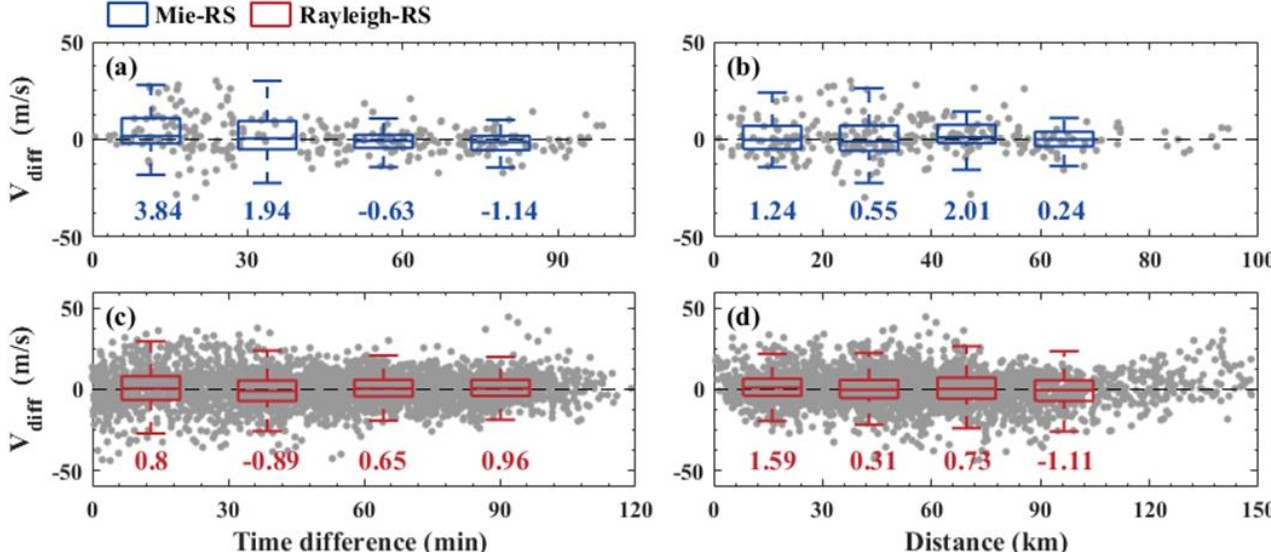

**Figure 4.** Difference between the Aeolus and RS HLOS as a function of (a, c) time difference and (b, d) distance. Blue, red box and gray dots indicate the Mie-cloudy, Rayleigh-clear winds and the corresponding sample points, respectively. The text labels represent the mean difference of the differences between Aeolus vs RS for each individual bin. The box size shows the upper margin, upper quartile, median, lower quartile, lower margin of the difference in each bin.


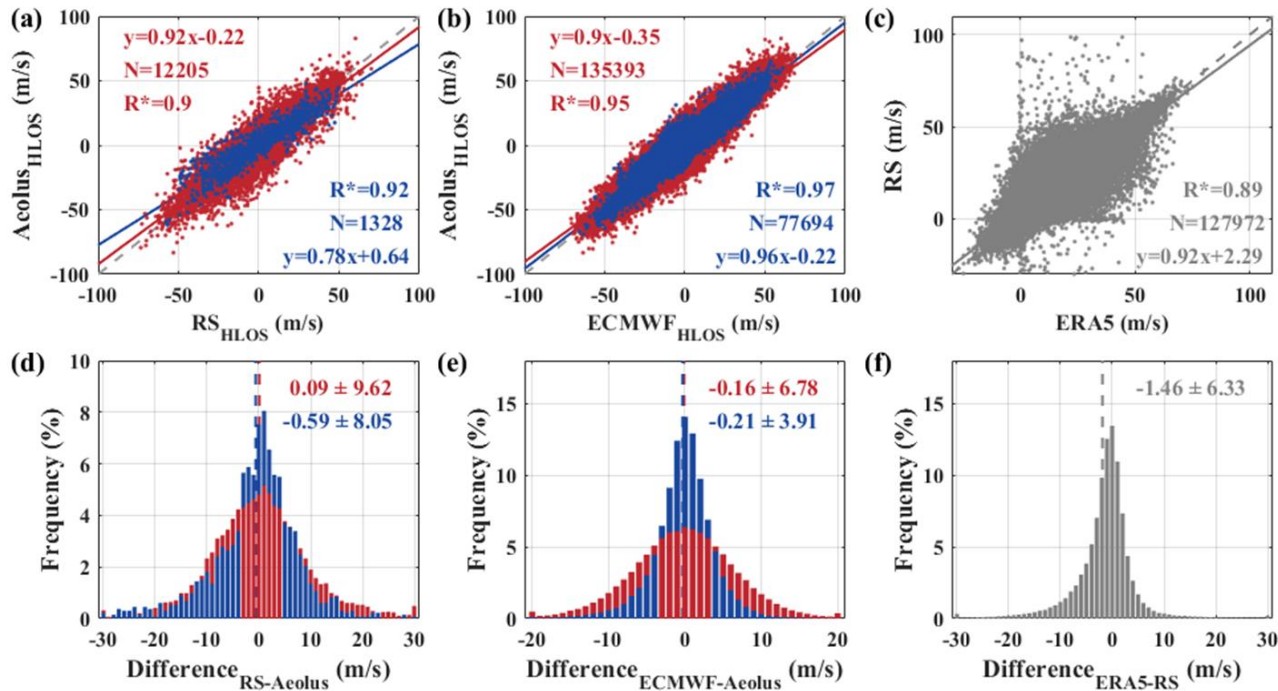

**Figure 5.** Scatterplots and histograms for (a, d) RS and Aeolus HLOS winds, (b,e) ECMWF and Aeolus HLOS winds and (c,f) ERA5-RS zonal wind components comparison results. Corresponding least-square line fits are indicated by the solid lines. The fit results are shown in the insets. The histogram plots show the mean and standard deviation in the insets. Blue and red indicate the Mie-cloudy and Rayleigh-clear winds, respectively. The 1:1 line is represented by the gray dashed line.

**Figure 6.** Geographic distribution of correlation coefficients between (a) Mie and RS, (b) Rayleigh and RS, (c) Mie and ECMWF, (d) Rayleigh and ECMWF and (e) RS and ERA5 winds. The black circles indicate the RS sites where the correlation coefficients between each data pair passed the statistic significance test (P<0.05).





**Figure 7.** Geographic distribution of averaged HLOS wind of Aeolus at (a, d) 850, (b, e) 500, and (c, f) 100 hPa and similar for RS zonal wind components at the different heights (resp. g, h and i). The ERA5 zonal wind component data is shown as the background values.

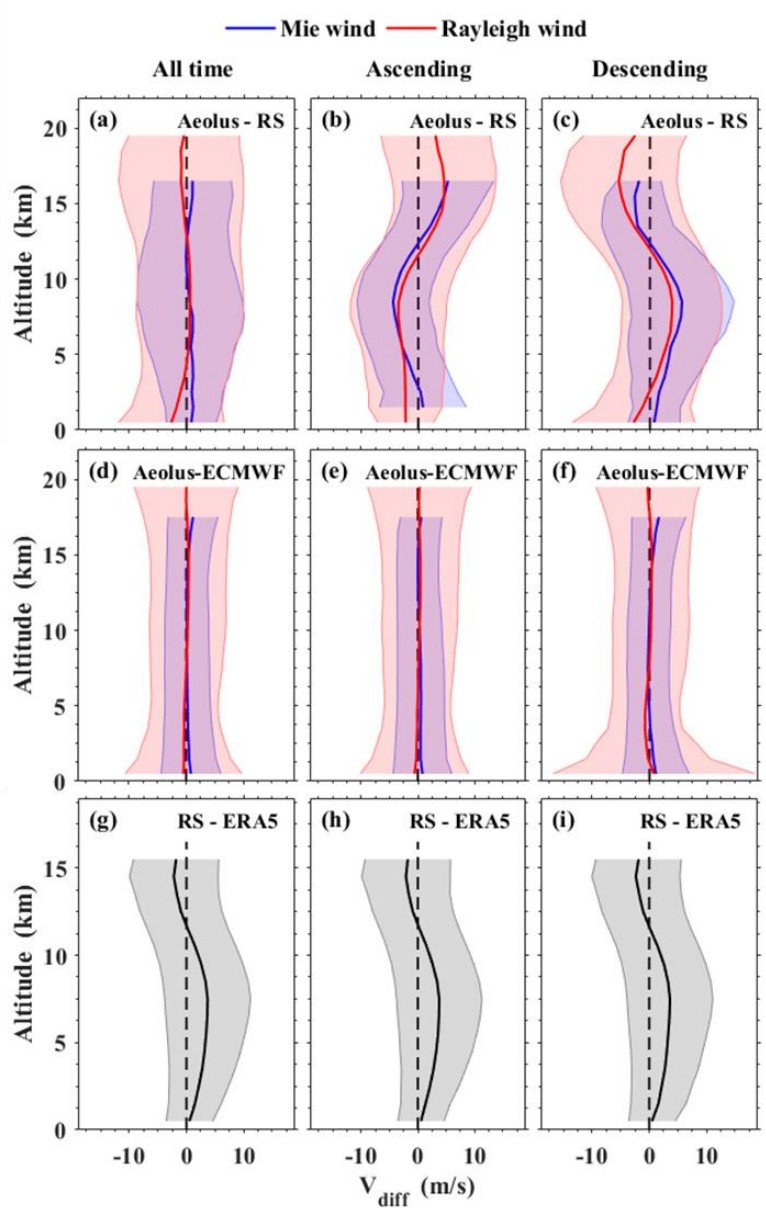

**Figure 8.** Vertical distributions of the wind HLOS difference between Aeolus, RS and ECMWF for (a, d) all time, (b, e) ascending, and (c, f) descending orbits and similar for RS and ERA5 zonal wind components at the different categories (resp. g, h and i)..

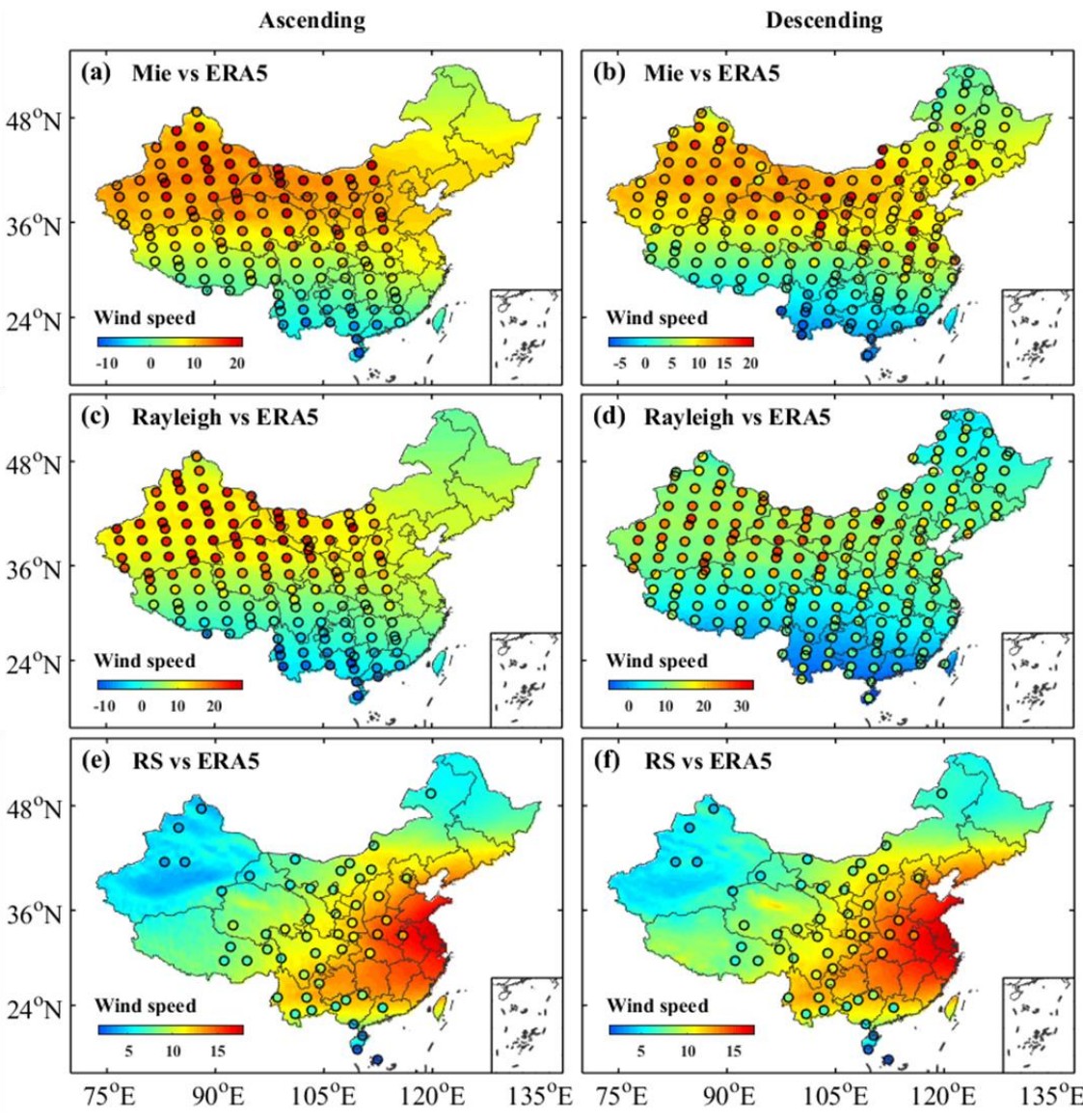

**Figure 9.** Geographic distribution of average HLOS wind of Aeolus during (a, c) ascending and (b, d) descending obits and similar for RS zonal wind components during ascending and descending orbits (resp. e and f). The ERA5 zonal wind component data (color shading) are shown as the background values.

**Figure 10.** Geographic distribution of HLOS difference between (a) Mie-RS, (b) Rayleigh-RS, (c) Mie-ECMWF, (d) Rayleigh-ECMWF wind speeds and similar for the difference between RS-ERA5 zonal winds (resp. e) during the ascending obits. The circles highlighted in black indicate the RS sites where the wind differences are statistically significant according to the Pearson's $x^2$ test.



**Figure 11.** Similar to Fig.10, but during the descending obits (or 12:00 UTC RS).