# Peer review of "Intercomparison of wind observations from ESA's satellite mission Aeolus, ERA5 reanalysis and radiosonde over China"

_Atmospheric Chemistry and Physics, 2021_

## Referee Comment (RC1)

Review – Aeolus – Liu et al
Special Issue: Aeolus data and their application (AMT/ACP/WCD inter-journal SI)

Scientific significant — 3 (good).  Aeolus intercomparisons are relevant.
Scientific quality — 2 (fair).  There are too many statistics repeated in the paper.
Presentation quality — 2 (fair).  The figures are not well-constructed.

1. Does the paper address relevant scientific questions within the scope of ACP?
   1. yes
2. Does the paper present novel concepts, ideas, tools, or data?
   1. Some new data
3. Are substantial conclusions reached?
   1. Not very substantial
4. Are the scientific methods and assumptions valid and clearly outlined?
   1. I am concerned that the processing of some of the data was in error.
5. Are the results sufficient to support the interpretations and conclusions?
   1. They are insufficient.
6. Is the description of experiments and calculations sufficiently complete and precise to allow their reproduction by fellow scientists (traceability of results)?
   1. Yes, it is traceable (notwithstanding a possible processing error.
7. Do the authors give proper credit to related work and clearly indicate their own new/original contribution?
   1. I liked the summary of the other Aeolus references.  The authors should find references on relative observation impact experiments (e.g., Buehner et al 2018, Mon. Wea. Rev.).
8. Does the title clearly reflect the contents of the paper?
   1. Yes
9. Does the abstract provide a concise and complete summary?
   1. Yes
10. Is the overall presentation well-structured and clear?
    1. Yes, it is well-written overall.
11. Is the language fluent and precise?
    1. Acceptable
12. Are mathematical formulae, symbols, abbreviations, and units correctly defined and used?

1. Yes

13. Should any parts of the paper (text, formulae, figures, tables) be clarified, reduced, combined, or eliminated?

      1. Many of the figures were not well-designed.

14. Are the number and quality of references appropriate?

      1. Yes

15. Is the amount and quality of supplementary material appropriate?

      1. Yes

General comments

Significant: The difference Aeolus-ERA mean zonal wind differences for ascending and descending show a significant negative bias on ascending (toward NW) passes and positive zonal wind bias on descending (toward SW) passes. Is there any corroboration of these results from any other studies? This is a critical result. As is discussed below regarding Fig. 8, it appears that the RS data may have their own bias, which would undercut the conclusions of this paper. I have a hard time concluding that the ECMWF data has such a zonal wind bias.

Fig.8 is potentially the most important result of the paper. Since ECMWF includes the Aeolus winds, but ERA5 does not, could one conclude that there is a bias with the RS winds? It appears that this is the case. How do the authors explain this issue? On p.13, L14-15, they suggest a possible ECMWF wind bias, but there is no other evidence that that is the case.
- I assume that Chinese RS data are used in both ECMWF and ERA5 analyses. Is this the case? The text should specify this clearly.
- The paper needs adding profiles for Aeolus winds (clear/cloudy/all) vs. ERA5. After all, Aeolus vs. ERA5 is shown in Fig. 9.

As noted below, I suspect a misprocessing, possibly vertical height assignment, for either the RS or ERA5 data.

**I understand that these statements will likely result in rejection of the manuscript. I urge the authors to either find the suspected error or more carefully support their logic and then, resubmit the manuscript. I thank the authors for their work in this intercomparison for the important Aeolus data even though I find these problems.**

The higher accuracy of Mie/cloudy winds than Rayleigh/clear winds are as expected – good. Can the authors provide correlation coefficients in Fig. 5 for other observation types? It's hard to tell what these numbers mean. I can only tell that they are from the same atmosphere at the same time with R* ~0.9.

Fig. 9 should use the same color scale for all graphics. This is confusing. Similarly, Fig. 7 should use the same color scale at least for the same vertical levels (850, 500, 100 hPa), but they don't. The colors in these graphics are confusing and should be applied uniformly.

Please remove national borders for areas of territorial disputes. The insert for the South China Sea area in the figures (1, 6, 7, 9, 10, 11) provide no scientific information whatsoever and should be removed.

Specific comments

p.2, L3-5. How is it known that this behavior is due to aerosols?

p.4, L1-2. The mean biases for A2D and Aeolus shown by Lux et al 2020 were *against the ECMWF model* – please correct.

The authors were not careful to ensure that the graphics all use similar scales. Instead, they simply used the range for each set of numbers rather than forcing a common range for the scales. Therefore, the colors mean different things for each plot. Fig. 5c does not use the same horizontal axis scale as 5a and 5b – please correct. Similarly, Fig. 5e does not use the same scale as Figs. 5d and 5f – again, please correct. Same problem for Fig. S1.

For all difference fields (Figs. 3, 7, 9, 10, 11), please use a white or gray color for differences close to zero, +/- 0.5 m/s for instance.

p.11, section 3. There is no need to repeat the statistics in this paragraph that are already obvious in the figures unless the authors want to draw some conclusion from those statistics. This paragraph can be shortened significantly.

p.11, L18. What data assimilation settings? Observation and background error values? If so, please say so.

Fig.7. What conclusions should the reader make from Fig. 7. There are some color differences the discussion on p.12-13 doesn't tell much except that the RS colors may very approximately match those from the other data sources.

Fig.8 other issues.
- Side issue: I suggest that the authors add the vertical profile of the number of Aeolus observations for Fig. 8. That may help solve the unexplained results in Fig. 8. Are there far fewer Aeolus observations for the 0-5km layer? I will guess that is the case.
- Is the altitude above sea level or above ground level? This should be added to the caption.
- What is the evidence that behavior in the 0-5km layer is affected by aerosols? This statement appears to be speculative.

- The paper needs adding profiles for Aeolus winds (clear/cloudy/all) vs. ERA5.  After all, Aeolus vs. ERA5 is shown in Fig. 9.

Fig.9.
- The stronger winds in RS than in ERA5 (Fig. 9e,f) possibly evident (very hard to visually average this) at least matches the vertical profile in Fig. 8e,f.   Why is it not possible that the RS values are in error?   Another possibility is an error in the vertical elevation matching using higher elevation in ERA5 or lower elevation in RS.   This would result in the ERA5 having stronger winds up to ~12km and lower winds above.

Fig.10.
- Please use the white or gray color for +/- 0.5 m/s.   Then revise the number of sites for which difference is negative, i.e., less than -0.5 m/s.
- Fig. 10e and its geographical pattern suggests the possibility of a misprocessing since the RS sites with larger differences are to the west, perhaps with higher elevation.

Technical corrections

p.1, L27.  It would be more accurate to say that the Aeolus winds observations were assimilated "into the ECMWF analysis", not "into the ECMWF winds" since the data assimilation is multivariate.
p.3, L18.  Aeolus misspelled.
p.6. ERA5 misspelled.
The references are considerably out of alphabetical order – this is a problem for reviewers.
Fig. 9.  'orbits', not 'obits'.  Same in p.13,L8.  Please spell-check the entire article and remove any 'obits'.

---

## Author Comment (AC1)

**Response to Reviewer #1's Comments**

*Response: We thank the anonymous reviewer for his/her comprehensive evaluation and thoughtful comments, which greatly improve the quality of our manuscript. We have made efforts to adequately address the reviewers' concern one by one. For clarity purpose, here we have listed the reviewer' comments in plain font, followed by our response in bold italics.*

Significant: The difference Aeolus-ERA mean zonal wind differences for ascending and descending show a significant negative bias on ascending (toward NW) passes and positive zonal wind bias on descending (toward SW) passes. Is there any corroboration of these results from any other studies? This is a critical result. As is discussed below regarding Fig. 8, it appears that the RS data may have their own bias, which would undercut the conclusions of this paper. I have a hard time concluding that the ECMWF data has such a zonal wind bias.

*Response: We appreciate your insightful comments. As you said, we made a mistake in the vertical height assignment of RS data. The reason is that we converted the height of RS data to the altitude above ground level. However, the height of Aeolus and ERA5 data are the altitude above sea level. This led to a series of wrong conclusions. We have corrected this error and provided new results, please see below.*

Fig.8 is potentially the most important result of the paper. Since ECMWF includes the Aeolus winds, but ERA5 does not, could one conclude that there is a bias with the RS winds? It appears that this is the case. How do the authors explain this issue? On p.13, L14-15, they suggest a possible ECMWF wind bias, but there is no other evidence that that is the case.

*Response: Good questions! Due to we made a mistake in the vertical height assignment of RS data, the previous results in Fig. 8 were wrong. We re-do the vertical height matching processing and provide new results. In addition, per your suggestion, we checked the number of matching samples at each height. To ensure the validity of the statistics, the comparison results with less than 20 matching samples were removed. The new Fig. 8 was show below. It found that the deviation in the vertical direction is significantly reduced. It is also worth mentioning that we*

*mistakenly believed that the large near-surface deviations were caused by aerosols.*
*In fact, it is caused by too few matching sample points.*

[Figure]

*Fig. 8*

- I assume that Chinese RS data are used in both ECMWF and ERA5 analyses. Is this the case? The text should specify this clearly.

*Response: Yes! The Chinese RS data are used in both ECMWF and ERA5 analyses. We have clarified it in the text.*

- The paper needs adding profiles for Aeolus winds (clear/cloudy/all) vs. ERA5. After all, Aeolus vs. ERA5 is shown in Fig. 9.

*Response: Due to the problem of vertical height matching, we did not add the profile comparison of Aeolus and ERA5. The Aeolus wind profile was a vertical resolution of 0.25 to 2 km in 0-20 km. The wind speed on each bin is calculated from the integral of the signal on this bin. However, the ERA5 data is a layered data, which has a total of 28 layers in the height range of 0–20 km. It means that for each Aeolus bins, the height interval of the bin is hard to be covered by ERA5 data. Therefore, we think that it is not appropriate to use the value of a certain layer to match the corresponding Aeolus bin value.*

*As for the comparison between Aeolus and ERA5 in Figure 9, these are actually average wind speeds within 0-20 km. The purpose of Figure 9 is to see if the Aeolus and ERA5 data are consistent in spatial distribution.*

As noted below, I suspect a misprocessing, possibly vertical height assignment, for either the RS or ERA5 data.

*Response: This is a very insightful comments. As you said, we made a mistake in the vertical height assignment of RS data. The height of Aeolus and ERA5 data are the altitude above sea level. However, we converted the height of RS data to the altitude above ground level, which resulted in the wrong results. We have corrected this error and provided new results.*

I understand that these statements will likely result in rejection of the manuscript. I urge the authors to either find the suspected error or more carefully support their logic and then, resubmit the manuscript. I thank the authors for their work in this intercomparison for the important Aeolus data even though I find these problems.

*Response: Thanks for your critical but valuable comments on our manuscript, which helps great in improving the quality of our manuscript. Please see the following point-by-point response to your comments.*

The higher accuracy of Mie/cloudy winds than Rayleigh/clear winds are as expected – good. Can the authors provide correlation coefficients in Fig. 5 for other observation types? It's hard to tell what these numbers mean. I can only tell that they are from the same atmosphere at the same time with R* ~0.9.

*Response: Per your suggestion, we modified Fig. 5, and updated Fig. 5 is shown as below:*

[Figure]

*Fig.5*

Fig. 9 should use the same color scale for all graphics. This is confusing. Similarly, Fig. 7 should use the same color scale at least for the same vertical levels (850, 500, 100 hPa), but they don't. The colors in these graphics are confusing and should be applied uniformly.

*Response: Amended as suggested. The same colour scale was used in Fig. 7 and 9.*

Please remove national borders for areas of territorial disputes. The insert for the South China Sea area in the figures (1, 6, 7, 9, 10, 11) provide no scientific information whatsoever and should be removed.

*Response: This suggestion has nothing to do with academic exchanges. Therefore, we don't think it is necessary to modify these pictures.*

p.2, L3-5. How is it known that this behavior is due to aerosols?

*Response: According to the new comparison results, we have deleted this wrong statement.*

p.4, L1-2. The mean biases for A2D and Aeolus shown by Lux et al 2020 were against the ECMWF model – please correct.

*Response: Amended as suggested.*

The authors were not careful to ensure that the graphics all use similar scales. Instead, they simply used the range for each set of numbers rather than forcing a common range for the scales. Therefore, the colors mean different things for each plot. Fig. 5c does not use the same horizontal axis scale as 5a and 5b, please correct. Similarly, Fig. 5e does not use the same scale as Figs. 5d and 5f, again, please correct. Same problem for Fig.S1.

*Response: Per your suggestion, we modified both Fig. 5 and Fig. S1.*

For all difference fields (Figs. 3, 7, 9, 10, 11), please use a white or gray color for differences close to zero, +/- 0.5 m/s for instance.

*Response: Per your suggestion, we modified the Fig. 10, 11 and S5. But for Fig. 3, 7 and 9, the color scale shows the wind speed value not the bias. Therefore, we did not modify the color scale of these three figures.*

p.11, section 3. There is no need to repeat the statistics in this paragraph that are already obvious in the figures unless the authors want to draw some conclusion from those statistics. This paragraph can be shortened significantly.

*Response: Good suggestion! We have rewritten this paragraph and deleted the redundant statistics*

p.11, L18. What data assimilation settings? Observation and background error values? If so, please say so.

*Response: The data assimilation settings mean the observation operator and expectations of 4D-var data assimilation, which is described in the TN (ECMWF TN 864, https://www.ecmwf.int/file/288329/download?token=y9cKewWP).*

*4D-var uses 3D spatial kernels for background (B) error representation, which spread the Aeolus observation (O) increments (O-B) in the ECMWF model domain. The kernels are based on the spread in an ensemble of forecasts and all observations in the temporal 4D-var window (12 hours) are considered to produce a ECMWF model trajectory that is consistent with all observations. The analysis weight of the Aeolus observations O depends on the local ratio of the estimated background and observation error covariances.*

*The Rayleigh winds have (variable) estimated errors associated with them, but these are inflated before data assimilation in accordance with the estimated Aeolus Rayleigh errors.*

*The Mie winds are on the 10-km scale and can be closely spaced horizontally. ECMWF follows arguments from Stoffelen et al. (2020) on observation density and spatial representativeness and found benefit in the forecasts by weight inflation of the Mie winds by a spatial representativeness error.*

*Per your suggestion, we clarified it the text.*

*Reference:*

*Stoffelen, A., Vogelzang, J., Marseille, G.-J.: High Resolution data assimilation guide. EUMETSAT NWP SAF Documentation, version 1.3, available: https://nwp-saf.eumetsat.int/site/download/documentation/scatterometer/reports/High_Resolution_Wind_Data_Assimilation_Guide_1.3.pdf, 2020.*

Fig.7. What conclusions should the reader make from Fig. 7. There are some color differences the discussion on p.12-13 doesn't tell much except that the RS colors may very approximately match those from the other data sources.

*Response: Figure 7 can be seen as a case study. Through comparison with ERA5 data, it shows the detection performance of Aeolus at different heights.*

Fig.8 other issues.

- Side issue: I suggest that the authors add the vertical profile of the number of Aeolus observations for Fig. 8. That may help solve the unexplained results in Fig. 8. Are there far fewer Aeolus observations for the 0-5km layer? I will guess that is the case.

*Response: Per your suggestion, we calculated the vertical profile of the number of Aeolus observations for Fig. 8, as shown in Fig. S4. It found that the number of Aeolus observations at near-surface is few. Therefore, the statistical results of the near surface are not credible. As you said, the large near-surface deviations were caused by too few matching sample points.*

[Figure]

**Fig. S4**

- Is the altitude above sea level or above ground level? This should be added to the caption.

*Response: The altitude is above sea level. We have added it to the caption.*

- What is the evidence that behavior in the 0-5km layer is affected by aerosols? This statement appears to be speculative.

*Response: According to the new comparison results, we have deleted this wrong statement.*

- The paper needs adding profiles for Aeolus winds (clear/cloudy/all) vs. ERA5. After all, Aeolus vs. ERA5 is shown in Fig. 9.

*Response: Due to the problem of vertical height matching, we did not add the profile comparison of Aeolus and ERA5. The Aeolus wind profile was a vertical resolution of 0.25 to 2 km in 0-20 km. The wind speed on each bin is calculated from the integral of the signal on this bin. However, the ERA5 data is a layered data, which has a total of 28 layers in the height range of 0–20 km. It means that for each Aeolus bins, the height interval of the bin is hard to be covered by ERA5 data. Therefore, we think that it is not appropriate to use the value of a certain layer to match the corresponding Aeolus bin value.*

*As for the comparison between Aeolus and ERA5 in Figure 9, these are actually average wind speeds within 0-20 km. The purpose of Figure 9 is to see if the Aeolus and ERA5 data are consistent in spatial distribution.*

Fig.9.

- The stronger winds in RS than in ERA5 (Fig. 9e,f) possibly evident (very hard to visually average this) at least matches the vertical profile in Fig. 8e,f. Why is it not possible that the RS values are in error? Another possibility is an error in the vertical elevation matching using higher elevation in ERA5 or lower elevation in RS. This would result in the ERA5 having stronger winds up to ~12km and lower winds above.

*Response: Good questions! As you said, we made a mistake in the vertical height assignment of RS data. It led to a series of wrong conclusions. We have corrected this error and provided new results. The new results show that the deviation between RS and ERA5 is very small.*

Fig.10. - Please use the white or gray color for +/- 0.5 m/s. Then revise the number of sites for which difference is negative, i.e., less than -0.5 m/s.

*Response: Amended as suggested.*

- Fig. 10e and its geographical pattern suggests the possibility of a misprocessing since the RS sites with larger differences are to the west, perhaps with higher elevation.

*Response: We have corrected this error and provided new results.*

p.1, L27. It would be more accurate to say that the Aeolus winds observations were assimilated "into the ECMWF analysis", not "into the ECMWF winds" since the data assimilation is multivariate.

*Response: Good suggestion! Amended as suggested.*

p.3, L18. Aeolus misspelled.

*Response: Amended as suggested.*

p.6. ERA5 misspelled.

*Response: Amended as suggested.*

The references are considerably out of alphabetical order – this is a problem for reviewers.

*Response: Amended as suggested.*

Fig. 9. 'orbits', not 'obits'. Same in p.13, L8. Please spell-check the entire article and remove any 'obits'.

*Response: Amended as suggested.*

---

## Author Comment (AC2)

**Response to Reviewer #2's Comments**

*Response: We greatly appreciated the reviewer's positive comments on our manuscript, which greatly improve the quality of our manuscript. We have made efforts to adequately address the reviewers' concern one by one. For clarity purpose, here we have listed the reviewer' comments in plain font, followed by our response in bold italics.*

The authors compare ESA's satellite Aeolus wind data with radiosonde winds in China in a period 20 April - 31 May. They also compare Aeolus wind fields with ERA5 wind fields. They use a fourth data set - "ECMWF wind fields" (need to be clarified, see below) as part of the Aeolus L2C data set - in a second period (July -Sept). Unfortunately, comparing all 4 data sets in an overlapping period was not possible. Numbers for correlations and mean differences are provided. Aeolus Rayleigh-clear winds and Mie-cloudy winds are considered separately. Conclusions are drawn by interpreting the various comparisons. They find that Aeolus winds are biased, and the bias is strongly different for ascending and descending orbits. They also find ERA 5 is biased over China. They find that a time difference criterion and a distance criterion does not seem to matter when they select Aeolus overpasses closest to the radiosonde start time and start location. The figures 1, 3, 4, 5, 7, 8, 9, 10, 11 are illustrating their work and supporting their conclusions. Major revisions are necessary before the paper allows the reader to understand what was done with which data and what conclusions can be drawn, and how relevant they are compared to what was already known.

*Response: Thanks for your valuable comments on our manuscript, which helps great in improving the quality of our manuscript. Please see the following point-by-point response to your comments.*

*In addition, we made a mistake in the vertical height assignment of RS data. The reason is that we converted the height of RS data to the altitude above ground level. However, the height of Aeolus and ERA5 data are the altitude above sea level. This led to a series of wrong conclusions. We have corrected this error and provided new results.*

Firstly, before publication, the authors need to describe exactly what the data are (versions), and who provided them. For instance, what is meant by "ECMWF data"? Were the background wind fields or the analysed wind fields used from the L2C product to compare to L2B? This makes a difference for the conclusion. Best support for their conclusion would be if they had used both background and analysis which would allow to illustrate how the change during data assimilation relates to Aeolus. Also, I assume L2C contains Aeolus obs error and data assimilation status flags, these would have been beneficial to consider in this paper. The L2B provided with the L2C data set contains Rayleigh winds only or also Mie, HLOS or components?

*Response: Good suggestion! Here, the analysed wind data from the L2C product was used to compare with L2B wind data. The L2C wind product adds two assimilation data modules on the basis of the L2B wind product: "L2C Mie Assimilation Product Confidence Data" and "L2C Rayleigh Assimilation Product Confidence Data". These two data modules are generated by the ECMWF model that assimilated the Aeolus observation data. It contains reference information such as the observation error, background error and data assimilation quality flags etc. For the entire L2C data product, it contains all L2B data and these two assimilation data modules.*

*Per your suggestion, we have added some descriptions in the section 2.3.*

Also, it would be important whether the Aeolus winds used in period 1 are comparable to period 2 - were they obtained with the same L2B processor? Are we comparing same Rayleigh / Mie winds?

*Response: Good question! According to the Aeolus official instructions ([https://earth.esa.int/eogateway/instruments/aladin/processor-releases](https://earth.esa.int/eogateway/instruments/aladin/processor-releases), last access: 22-06-2021), there are three processor releases: Baseline 12 (26 May 2021 – present), Baseline 11 (8 Oct 2020 – 26 May 2021) and Baseline 10 (20 Apr – 8 Oct 2020). In this study, the L2B data were from 20 April 2020 to 30 September 2020. Therefore, the L2B processor release during this period should be "Baseline 10".*

*In addition, by consulting with Dr. Stoffelen, A., we learn that generation of AUX_TEL file needed to perform the telescope temperature bias correction has changed from once to twice per day (based on the previous 24 hours of data) at 10-Aug-2020. This should give a small quality improvement. Therefore, we think that the Aeolus winds used in period 1 are similar to period 2.*

*To dispel readers' doubts, we have added a description in the section 2.1.*

Rayleigh and Mie winds cover different vertical ranges. Comparing both should take this into consideration. Where height dependency is considered (e.g. as done in Fig 7 and 8) conclusions can be drawn more easily.

*Response: Good suggestion! Due to we made a mistake in the vertical height assignment of RS data, the previous results in Fig. 8 were wrong. We re-do the vertical height matching processing and provide new results. The height dependency is also considered. The new results indicate that the deviation in the vertical direction is significantly reduced.*

Second issue before publication, the authors need to describe in a reproducible manner their data processing. Figure 2 is not useful. The two periods, and each data set should be described separately, Figure 2 is in contradiction to the text. They need to state for each comparison, which data were (automatically) excluded, as this determines their resulting means and correlation coefficients. Also, in section 2.5 the wind components are discussed. Most discussion in the paper refers to the horizontal line-of-sight (HLOS) wind. It is not clear, where Aeolus wind components are needed and which numbers refer to wind speed or wind components or HLOS winds.

*Response: Per your suggestion, we described the data processing process in two periods. The first period is from 20 April to 30 September 2020 for the comparison between Aeolus and RS data. Another period is from 9 July to 30 September 2020 for the Aeolus-ECMWF and RS-ERA5 comparison. The new Fig. 2 was shown below.*

*In addition, for the Aeolus-ECMWF and RS-ERA5 comparison, the wind data were both converted to Aeolus horizontal line-of-sight (HLOS) wind. Only for the comparison of RS and ERA5 data, the wind data were converted to the zonal wind component. We have clarified it in section 2.5.*

[Figure]

**Fig. 2**

Third issue, conclusions from the various comparisons have to be discussed in a scientifically rigorous manner. What can and what cannot be concluded from the 2 periods? Actually ERA5 should be available in both periods. However, known seasonal dependency of Aeolus biases might limit the option of drawing conclusions from the spring and autumn period ignoring their different season. Also, the known dependency on topography is ignored here but might matter (compare Fig. 7, 850 hPa)

*Response: Good questions! Due to we made a mistake in the vertical height assignment of RS data, most of the previous results were wrong. We re-do the vertical height matching processing and provide new results.*

Fourth major recommendation concerns bringing the findings of the paper into perspective with what is known from other literature, e.g., bias of Aeolus (ascending/descending) was known before (simple google search brought me to https://doi.org/10.5194/amt-2020-404), and so are ERA5 biases over complex topography. In the abstract it is concluded that the findings give sufficient information to apply Aeolus wind products in numerical weather prediction in China. This surely might be a valid point, but needs a criterion what is meant by "sufficient information". Are both Rayleigh and Mie winds considered useful, or one more than the other, useful

always or under certain circumstances? How does the size of differences between data sets compare to other literature? Discussing known literature will help to illustrate the added value of this paper, which is studying the region of China in detail.

*Response: Per your suggestion, we add some discussion in the text.*

*"Khaykin et al. (2020) also analyzed one wind profile of Aeolus with the Doppler lidar and found a good agreement between the two measurements, but below 5 km above ground level, a stronger deviation was observed, which was likely caused by horizontal heterogeneity of the atmosphere."*

*"Previous study also indicates that there are differences in bias between the ascending and descending orbit phase, which mainly occur for the Rayleigh channel in late summer and autumn (Martin et al., 2021)"*

*"The comparison results obtained in this study, by and large, agree well with most of validation work against Aeolus wind products, although the data sources and regions of interest vary a lot. For instance, Baars et al. (2020) revealed that the random errors were about 4 and 2.2 m/s for Rayleigh-clear and Mie-cloudy wind, respectively, by utilizing the RV Polarstern cruise from Bremerhaven to Cape Town. Lux et al. (2020) compared the Aeolus Rayleigh-clear wind observations to winds measured with the airborne demonstrator and the ECMWF model in central Europe. They reported a bias of 1.6 (2.53) m/s with random errors of 2.5 (3.57) m/s for the comparison against the ECMWF model (airborne demonstrator). In a recent comparison analysis based on a combination of Aeolus, RS and numerical weather prediction model on a global scale, the mean absolute bias is found to be approximately 1.8–2.3 m/s for the Rayleigh winds and 1.3–1.9 m/s for the Mie winds (Martin et al., 2021)."*

*References:*
*Benjamin, S. G., Schwartz, B. E., Szoke, E. J., and Koch, S. E.: The value of wind profiler data in US weather forecasting. Bulletin of the American Meteorological Society, 85(12), 1871-1886, 2004.*

*Baars, H., Herzog, A., Heese, B., Ohneiser, K., Hanbuch, K., Hofer, J., Yin, Z., Engelmann, R., and Wandinger, U.: Validation of Aeolus wind products above the Atlantic Ocean, Atmos. Meas. Tech., 13, 6007–6024, https://doi.org/10.5194/amt-13-6007-2020, 2020.*

*Khaykin, S. M., Hauchecorne, A., Wing, R., Keckhut, P., Godin-Beekmann, S., Porteneuve, J., Mariscal, J.-F., and Schmitt, J.: Doppler lidar at Observatoire de*

*Haute-Provence for wind profiling up to 75 km altitude: performance evaluation and observations, Atmos. Meas. Tech., 13, 1501–1516, https://doi.org/10.5194/amt-13-1501-2020, 2020.*

*Martin, A., Weissmann, M., Reitebuch, O., Rennie, M., Geiß, A., and Cress, A.: Validation of Aeolus winds using radiosonde observations and numerical weather prediction model equivalents, Atmos. Meas. Tech., 14, 2167–2183, https://doi.org/10.5194/amt-14-2167-2021, 2021.*

Several of the experienced co-authors should be able to rewrite this paper in a more scientifically stringent manner.

*Response: Per your suggestion, we rephrased most of section in this revision in a more scientifically stringent manner.*